# Hesperidin Is a Potential Inhibitor against SARS-CoV-2 Infection

**DOI:** 10.3390/nu13082800

**Published:** 2021-08-16

**Authors:** Fang-Ju Cheng, Thanh-Kieu Huynh, Chia-Shin Yang, Dai-Wei Hu, Yi-Cheng Shen, Chih-Yen Tu, Yang-Chang Wu, Chih-Hsin Tang, Wei-Chien Huang, Yeh Chen, Chien-Yi Ho

**Affiliations:** 1Drug Development Center, China Medical University, Taichung 404, Taiwan; fangju27@gmail.com; 2Graduate Institute of Biomedical Science, China Medical University, Taichung 404, Taiwan; huynhkieuthanh@gmail.com (T.-K.H.); hudebby1024@gmail.com (D.-W.H.); greywolf0127@gmail.com (Y.-C.S.); chtang@mail.cmu.edu.tw (C.-H.T.); 3Institute of New Drug Development, China Medical University, Taichung 404, Taiwan; xin740103@hotmail.com; 4Department of Internal Medicine, Division of Pulmonary and Critical Care Medicine, China Medical University Hospital, Taichung 404, Taiwan; chesttu@gmail.com; 5School of Medicine, China Medical University, Taichung 404, Taiwan; 6Chinese Medicine Research and Development Center, China Medical University Hospital, Taichung 404, Taiwan; yachwu@mail.cmu.edu.tw; 7Graduate Institute of Integrated Medicine, College of Chinese Medicine, China Medical University, Taichung 404, Taiwan; 8Department of Medical Laboratory Science and Biotechnology, College of Medical and Health Science, Asia University, Taichung 404, Taiwan; 9Research Center for Cancer Biology and Center for Molecular Medicine, China Medical University, Taichung 404, Taiwan; 10Department of Biomedical Imaging and Radiological Science, China Medical University, Taichung 404, Taiwan; 11Division of Family Medicine, Physical Examination Center, Department of Medical Research, China Medical University Hsinchu Hospital, Hsinchu 302, Taiwan

**Keywords:** hesperidin, TMPRSS2, ACE2, SARS-CoV-2, D614G, 501Y.v2

## Abstract

Hesperidin (HD) is a common flavanone glycoside isolated from citrus fruits and possesses great potential for cardiovascular protection. Hesperetin (HT) is an aglycone metabolite of HD with high bioavailability. Through the docking simulation, HD and HT have shown their potential to bind to two cellular proteins: transmembrane serine protease 2 (TMPRSS2) and angiotensin-converting enzyme 2 (ACE2), which are required for the cellular entry of severe acute respiratory syndrome coronavirus 2 (SARS-CoV-2). Our results further found that HT and HD suppressed the infection of VeroE6 cells using lentiviral-based pseudo-particles with wild types and variants of SARS-CoV-2 with spike (S) proteins, by blocking the interaction between the S protein and cellular receptor ACE2 and reducing ACE2 and TMPRSS2 expression. In summary, hesperidin is a potential TMPRSS2 inhibitor for the reduction of the SARS-CoV-2 infection.

## 1. Introduction

The severe acute respiratory syndrome coronavirus 2 (SARS-CoV-2), a single-stranded and positive-sense RNA virus in the betacoronavirus family, caused the global pandemic of coronavirus disease in 2019 (COVID-19) [1,2]. Compared to SARS-CoV and the Middle East respiratory syndrome coronavirus (MERS-CoV), SARS-CoV-2 exhibited a higher rate of human-to-human transmission and has threatened global health with a high mortality rate [3]. The development of an effective strategy in controlling the rapid spread of COVID-19 is an urgent issue.

The cell entry of SARS-CoV-2 depends on the binding of the viral spike (S) protein to the cellular receptor angiotensin-converting enzyme2 (ACE2). Through the binding between the S1 subunit and ACE2, the S protein is cleaved into the domains of S1 and S2 by host transmembrane serine protease 2 (TMPRSS2) to expose the fusion peptide on S2 for the subsequent membrane fusion and cellular entry of viral RNA [4,5]. After the cellular entry, both the viral 3-chymotrypsin-like protease (3CLpro; also called Mpro) and papain-like protease (PLpro) further cleave and process the viral polyproteins for the production of four essential structural proteins: the S protein, nucleocapsid (N) protein, membrane (M) protein, and envelope (E) protein, which are needed to compose a complete viral particle [6,7]. To date, COVID-19 has infected over 160 million people, resulting in 3 million deaths. It is imperative to search for potential and protective treatments against SARS-CoV-2 infection. For the development of preventive or therapeutic approaches against SARS-CoV-2 infection, the critical proteins involved in the cellular attachment and replication of SARS-CoV-2 are considered as effective targets [5,8,9,10].

Food has potential benefits to block COVID-19 by modulating the immune system or defending oxidative stress in response to virus infection [11]. Hesperidin (HD; 3,5,7-trihydroflavanone 7-rhamnoglucoside), a major functional flavanone in flavonoids, can be isolated from lemons and other citrus fruits [12], from which rhamnose sugar can also be removed by glycosyl hydrolases to form hesperetin (HT; 3′, 5,7,-trihydroxy-4′-methoxyflavanone) [13]. Due to the removal of rhamnose sugar, HT has a higher bioavailability compared to HD and can be absorbed directly into the small intestine [13,14]. It has been documented that HD possesses several pharmacological effects, primarily the promotion of anti-oxidation, suppression of pro-inflammatory cytokine production, and repression of cancer cell growth [15]. In addition, HD attenuated the influenza A virus (H1N1)-induced secretion of pro-inflammatory cytokines, contributing to the improvement of pulmonary function [16]. More recently, HD has been shown to possess the anti-SARS-CoV-2 infection activity in an in vitro cell line model [17], and is predicted to bind to viral S and 3CLpro proteins of SARS-CoV-2 and cellular ACE2 of host cells in the molecular simulation analysis [18,19], implying the anti-SARS CoV-2 potential of HD. However, there is no substantial evidence to address the activity of HD/HT against SARS-CoV-2 infection. In this study, we addressed the molecular mechanisms underlying the anti-SARS-CoV-2 infection of HD and HT.

## 2. Materials and Methods

### 2.1. Cell Lines and Cell Culture

The human Beas 2B lung cell line was grown in Dulbecco’s Modified Eagle Medium (DMEM) with low glucose (Gibco, Waltham, MA, USA, Cat. No. 31600083); the human NCI-H460 lung cancer cell line was grown in RPMI 1640 medium (Gibco, Waltham, MA, USA, Cat. No. 31800089), and monkey VeroE6 kidney cells were cultured in DMEM medium with GlutaMax supplement (Gibco, Waltham, MA, USA, Cat. No. 31966021). All cell lines were cultured in the presence of 10% FBS, 1% streptomycin (100 μg/mL), 1.5 g/L sodium bicarbonate, 1 mM sodium pyruvate, and HEPES (25 mM) at 37 °C in a humidified 5% CO_2_/95% incubator.

### 2.2. Measurement of Luciferase Intensity of SARS-CoV-2 Spike Protein-Pseudotyped Lentiviral Particles

VSV-G pseudotyped lentivirus and the virus particle pseudotyped (Vpp) of SARS-CoV-2 Spike protein SARS-CoV-2-S) with luciferase were obtained from RNA Technology Platform and Gene Manipulation Core, Academia Sinica in Taiwan. The veroE6 cell line was treated with 100 μM of hesperetin and hesperidin for 2 days and then was infected with nCoV-2 pseudovirus. After 24 h, the infected cells were lysed with One-Glo^TM^ Luciferase assay buffer (Promega, Madison, WI, USA, Cat. No. E6120), and luciferase activity was measured using a luminometer.

### 2.3. Measurement of 3-(4,5-Dimethylthiazol-2-yl)-2,5-diphenyltetrazolium Bromide (MTT)

The cell viability was measured according to the manufacturer’s protocol. VeroE6 cell line (8000 cells) in 96-well plates were treated with HT and HD in a dose-dependent manner for 2 days and then subjected to MTT assay (Sigma-Aldrich, Cat. No. M2003).

### 2.4. Docking-Pose Prediction

For the first step of docking simulation, the crystal structures of TMPRSS2 (PDB code: 1Z8A, accessed date: 14 March 2006), ACE2 (PDB code: 3D0G, accessed date: 8 July 2008), PLpro (PDB code: 3E9S, 7 October 2008), and 3CLpro (PDB code: 3AW0, 14 December 2011) were received from Protein Data Bank (PDB, https://www.rcsb.org/). The structures of HT (PDB code: 5JDC) and HD (PDB code: 6CCF) are also from PDB. Discovery Studio, a docking software using a genetic algorithm, was employed to simulate the automated docking of HT and HD with the catalytic sites of proteins as mentioned above.

### 2.5. FRET-Based Enzyme Activity Assay

The effects of HT and HD on the interaction between SARS-CoV-2 Spike S1 and human ACE2 were measured by TR-FRET assay according to the manufacturer’s protocol (BPS Bioscience, Inc., San Diego, CA, USA, Catalog #79949-1). Briefly, ACE2 and Spike S1 proteins, with or without 60 μM tested compounds, were incubated at room temperature for 1 h. TR-FRET signals were recorded by detecting the emission at a wavelength 620 or 665 nm with excitation at a wavelength 340 nm. HT and HD almost had no effect on the fluorescence of individual proteins. The data were normalized with the effect of tested inhibitors on the fluorescence of ACE2. In the examination of the inhibitory effects of HT and HD on protease activity of human TMPRSS2, the reaction mixture containing 15 μg/mL recombinant protein (Creative Biomart Inc., Shirley, NY, USA, Cat. No. TMPRSS2-1856H) and 60 μM HT or HD in assay buffer (25 mM Tris 8.0, 150 mM NaCl) was pre-incubated at room temperature for 30 min. The reaction was initiated by the addition of 20 μM fluorescent protein substrate. Substrate cleavage was monitored continuously for 6 h by detecting mNeonGreen fluorescence (excitation: 506 nm/emission: 536 nm) using Synergy™ H1 hybrid multi-mode microplate reader (BioTek Instruments, Inc., Winooski, VT, USA). The first 1 h of the reaction was used to calculate initial velocity (V_0_). The initial velocity with each compound was calculated and normalized to DMSO control. The effects of HT and HD on SARS-CoV-2 Papain-like Protease (PLpro) were examined by using Papain-like Protease Assay Kit (BPS Bioscience, San Diego, CA, USA, Catalog #79995-2) according to the manufacturer’s protocol. Briefly, PLpro was incubated with 60 μM HT or HD for 1 h at 37 °C. The peptide substrate was then added to start the reaction. The fluorescence signal was monitored continuously for 1 h by detecting emission at a wavelength of 460 nm with excitation at a wavelength of 360 nm. The preparation of recombinant SARS-CoV-2 3CLpro and the measurement of its enzyme activity assay were described previously [20]. Briefly, 60 μM HT or HD was pre-incubated with SARS-CoV-2 3CLpro for 30 min at room temperature. The reaction was started by the addition of 20 μM fluorescent protein substrate. The fluorescent signal (Ex/Em: 434 nm/474 nm) was continuously monitored for 1 h. The first 15 min of the reaction was used to calculate initial velocity (V_0_) and was normalized to DMSO control. All the data were shown as mean ± SEM from three independent experiments performed in at least three replicates.

### 2.6. Western Blotting

Cell lines treated with HT and HD in a dose-dependent manner for 2 days were lysed in RIPA buffer containing phosphatase and protease inhibitors. The protein lysates were separated by SDS-PAGE and were then transferred to PVDF membranes. The membranes were blocked in 5% milk in TBST buffer (TBS with 0.1% Tween 20), and were incubated with primary antibodies, including ACE2 (Genetex, Hsinchu, Taiwan, Cat. No. GTX101395), TMPRSS2 (Santa Cruz, CA, USA, Cat. No. sc-515727), and β-actin (Sigma-Aldrich, Darmstadt, Germany, Cat. No. A2228), for overnight at 4 °C followed by incubation with HRP-conjugated second antibody for 1 h at room temperature. After washing with TBST buffer, the immunoreactive signals were visualized by using enhanced chemiluminescence with ECL reagent.

### 2.7. Statistical Analysis

Data were shown as the mean ± standard error of the mean (SEM). A one-way ANOVA was used for most comparisons. A *p*-value < 0.05 was considered statistically significant.

## 3. Results

### 3.1. Molecular Docking Reveals HT and HD as Potential Multiple-Target Inhibitors against COVID-19 

To test whether HT and HD possess potential activity against SARS-CoV-2 infection, we performed a molecular docking simulation to predict the binding affinity of these two compounds to cellular proteins involved in the cellular entry of SARS-CoV-2 and to viral proteases of SARS-CoV2. The results showed that ACE2 (Figure 1A–F) and TMPRSS2 (Figure 1G–L), the cellular proteins involved in the entry of SARS-CoV2, could interact with HT and HD, and that the energy values of ACE2-HT, ACE2-HD, TMPRSS2-HT, and TMPRSS2-HD were −34.81, −1.65, −30.56, and −7.2 kCal/mol, respectively (Table 1). In addition, these two compounds also yielded an interaction with PLpro (Figure 2A–F) and 3CLpro (Figure 2G–L), the viral proteins involved in the replication of SARS-CoV2. The predicted energy values of PLpro-HT, PLpro-HD, 3CLpro-HT, and 3CLpro-HD were −13.69, 17, −17.94, and 6.21 kCal/mol, respectively (Table 1). Additionally, the binding intensity of HT to PLpro and 3CLpro was higher than that of HD (Table 1). Taken together, HT and HD were potentially natural agents against COVID-19 infection by interfering with the cellular entry and virus replication. 

### 3.2. HT Suppresses the Interaction between ACE2 and the Spike Protein of SARS-CoV-2 In Vitro

To further confirm the inhibitory effects of HT and HD on SARS-CoV-2 infection, we performed a FRET assay to examine the binding affinity between human receptor ACE2 and the S protein, as well as the enzymatic activities of TMPRSS2, PLpro, and 3CLpro in the presence of HT and HD. The result displayed that the treatment with HT but not HD reduced the binding activity between ACE2 and the S protein (Figure 3A). The result of the docking simulation also showed that HT entered the pocket of human ACE2 bound to the S protein (Appendix A). In addition, HT and HD only slightly decreased the enzyme activity of TMPRSS2 (Figure 3B) but did not influence the enzyme activity of PLpro (Figure 3C) and 3CLpro (Figure 3D), suggesting that HT had the potential role of blocking the cellular entry of SARS-CoV-2 via impeding the binding of human receptor ACE2 with the S protein.

### 3.3. HT and HD Downregulated the Protein Expression of ACE2 and TMPRSS2 in Normal and Malignant Lung Cells

In order to examine the cytotoxicity of HT and HD, we performed MTT assays to test their effect on the cell viability of VeroE6 cells in a dose-dependent manner. The IC50 values of HT and HD in VeroE6 cell were 1491 and 1435 μM, respectively (Appendix A), indicating their low toxicity to cells. We next examined whether HT and HD influence the cellular protein expressions of ACE2 and TMPRSS2 in Western blot analysis. In Figure 4, treatments with HT and HD repressed the protein expressions of ACE2 and TMPRSS2 in normal lung epithelial Beas 2B cell (Figure 4A) and in H460 lung cancer cells (Figure 4B), suggesting that HT and HD not only disrupted the interaction of ACE2 and SARS-CoV-2 S protein but also inhibited the protein expressions of ACE2 and TMPRSS2 for the reduction of SARS-CoV-2 infection. As shown in Appendix A, we unexpectedly found that both HT and HD increased but did not decrease the mRNA levels of ACE2 and TMPRSS2, suggesting a post-transcriptional downregulation of the two proteins by HT and HD. Interestingly, heat shock protein 70/90 (HSP70/90) was found to mediate the protein stabilization of ACE2 and thereby maintained the cell entry mechanism of SARS-CoV-2 [21]. HSP90 was reported to be significantly downregulated in the hesperidin-treated cells [22]. These findings raise the possibility that HT and HD functions as HSP70/90 inhibitors to cause the ACE2 and TMPRSS2 protein downregulation despite the induction of their mRNA expressions (Appendix A).

### 3.4. HT and HD Block the Cellular Entry of Vpp of SARS-CoV-2 Spike Protein (SARS-CoV-2-S)

Next, we verified the potential of HT and HD in suppressing the cellular entry of SARS-CoV-2. The VeroE6 monkey kidney cell line was infected with the Vpp of SARS-CoV-2-S followed by the pretreatments with HT and HD for 2 days due to their inhibitory effects on ACE2 and TMPRSS2 expressions. We found that treatments with HT and HD significantly impaired the infection of SARS-CoV-2-S with the Vpp (Figure 5B), but not VSVG pseudotyped lentivirus (Figure 5A). The globally uncontrolled transmission of SARS-CoV2 was due to the viral evolution. Starting from April 2020, the predominant strains of SARS-CoV2 were D614G (substitution of aspartate (D) to glycine (G) at site 614 in S protein) and 501Y.v2 (also called B.1.351; the simultaneous mutation of D614G and N501Y in the S protein) [23,24]. Compared to the wild-type S protein of SARS-CoV-2, both of these variants showed more robust binding activities to ACE2 to increase the efficacy of virus replication and transmission in host cells [25]. Treatments with HT and HD also dramatically diminished the cellular entry of the VPP of SARS-CoV-2-S, D614G and 501Y.v2 strains (Figure 5B,C), without affecting the viability of VeroE6 cells (Appendix A).

Furthermore, we treated VeroE6 cells with these compounds in the short term, before or after virus infection (as illustrated in Appendix A), to demonstrate that the blockage of ACE/S protein interaction mediated the anti-SARS-CoV-2 infection activity of HT. As shown in Appendix A, the cell entry of the SARS-CoV-2 pseudovirus with wild types and variants of the S protein was reduced by adding HT (red) at 2 h pre-infection or during infection, but not post-infection, of pseudovirus. Interestingly, the treatments with HD (blue) in the short term did not suppress the infection of SARS-CoV-2 pseudovirus, which was consistent with its inability to suppress the interaction between ACE2 and S protein (Figure 3A) These results suggest HT and HD as potential agents against infection with SARS-CoV2 and its mutant strains.

## 4. Discussion

To date, the COVID-19 pandemic still cannot be controlled, even though many therapeutic strategies and vaccines have been developed. Similar to other RNA viruses, the genetic diversity of SARS-CoV-2 driven by high-random mutation and recombination enables this virus to increase the recognition of human cellular receptors, virus replication, and the higher rate of widespread infection [24,26,27]. However, these variants of SARS-CoV-2, such as the predominant strains of D614G and 501Y.v2, impair the neutralization by the vaccine-induced immunity [23,28]. In this study, we found that HT and HD are able to dramatically inhibit the cellular entry of Vpp of SARS-CoV-2 variants (Figure 5). HD has been considered as a favorable adjuvant for vaccines to prevent lung injury by promoting pro-inflammatory cytokines secretion and cell-autonomous immunity against influenza A (H1N1) infection [16,29,30]. Taken together, HT and HD may show benefits in fighting the threat of the COVID-19 pandemic.

There has been a wide prevalence of SARS-CoV-2 infection, and specific medicines for COVID-19 remain unavailable. Therefore, the components of SARS-CoV-2 and infection procedure are potential targets to screen for the pre-existing or marketed drugs which may possess preventive or therapeutic activity against SARS-CoV2 infection. Some potential medicines have been found to suppress SARS-CoV-2 infection in in vitro and animal studies [31]. However, few medicines have been proven to eradicate SARS-CoV-2 infection effectively in clinical trials. The use of chloroquine and hydroxychloquine, which inhibit the cellular entry, was revoked due to the high risk of mortality [32]. Camostat and Nafamostat, the synthetic inhibitors for TMPRSS2, also suppressed the cellular entry of SARS-CoV-2 but caused severe bleeding [33]. Our findings identified that HT and HD hindered the interaction between the S protein of SARS-CoV-2, hosted the cellular receptor ACE2 and downregulated the protein expression of ACE2 and TMPRSS2, thereby suppressing the infection with Vpp of SARS-CoV-2-S. In addition, the administration of HD at 500 mg/kg did not cause any abnormalities in the animal model, indicating a good safety profile. The median lethal dose of HD is 4837.5 mg/kg [34]. Therefore, HD could be considered as an effective and natural compound to fight SARS-CoV-2 infection.

*Poncirus trifoliata* (L.) *Raf.* (also called bitter orange fruit) belongs to the member of Rutaceae family and is closely related to *Citrus trifoliata*. This fruit contains various phytotherapeutic activities, which depend on its maturity, to alleviate symptoms in disorders. Poncirus fructus, the dry form of immature fruit of *Poncirus trifoliata* (L.) *Raf.*, is commonly known as a herbal medicine in East Asia for the dysfunction of the digestive system. The mature fruits demonstrated anti-cancer and anti-inflammatory activities [35]. Moreover, the seed extract from *Poncirus trifoliata* (L.) *Raf.* possessed the antiviral activity via the suppression of the cellular endocytosis of the oseltamivir-resistant influenza virus [36]. HD is one of the predominant phytochemicals found in *Poncirus trifoliata* (L.) *Raf* [37,38] and possesses the antioxidant, anti-inflammation, and anti-tumor properties [15]. These findings and our study support that HD isolated from *Poncirus trifoliata* (L.) *Raf.* could be considered as a potential agent to prevent SARS-CoV-2 infection.

Several review articles have predicted the anti-SARS-CoV-2 infection activity of the flavonoid family, including hesperidin [18,19]. Kandeil et al. cleanly demonstrated the inhibitory effect of hesperidin on the viral replication of SARS-CoV-2 at the early stage of virus infection [17]. Unfortunately, there is no direct evidence showing that the molecular mechanism of anti-SARS-CoV-2 activity is derived from hesperidin. It is also unknown whether hesperidin possesses anti-SARS-CoV-2 activity through the influence of the cellular components involved in SARS-CoV-2 infection. In this study, we demonstrated that hesperidin and its aglycone metabolite hesperetin repressed the protein expression of ACE2 and TMPRSS2 in lung cells, impeding the cell entry of the SARS-CoV-2 pseudo-virus. But hesperidin cannot directly decrease the activities of viral proteases, including PLpro and Mpro, in the enzyme activity assays (Figure 3C,D) even though the binding activity of hesperidin to these enzymes was predicted in the molecular docking analysis in the previous studies of [18,19], and this study (Figure 2G–L; Table 1). The reliability of protein structure and the environment of the binding site used for the ligand–protein complex docking assays would determine the prediction accuracy [39]. The molecular dynamic simulation would also be required to validate the predictions from molecular docking [39].

## 5. Conclusions

Currently, no specific therapy can significantly inhibit SARS-CoV-2 infection and help to prevent the COVID-19 pandemic in many parts of the world. Several vaccines were developed and approved by FDA to prevent SARS-CoV-2 infection and effectively suppressed the incidence of COVID-19 [40,41]. However, SARS-CoV-2 variants escaped from the inhibition by neutralizing antibodies [28]. Exploring the promising antiviral agents remains essential for the termination of SARS-CoV-2 spreading. Hesperidin is enriched in citrus fruits, which are common traditional medicines in Asia. In this study, we demonstrated that hesperidin and its aglycone, hesperetin, might provide benefits in fighting COVID-19 via blocking the binding of the S protein of SARS-CoV-2 to the human cellular receptor ACE2 and reducing the protein expression of ACE2 and TMPRSS2. These effects significantly suppress the cellular entry of the SARS-CoV-2 variant regardless of the mutation of the S protein. Therefore, hesperidin could be used as a potential prophylactic treatment against COVID-19.

## Figures and Tables

**Figure 1 nutrients-13-02800-f001:**
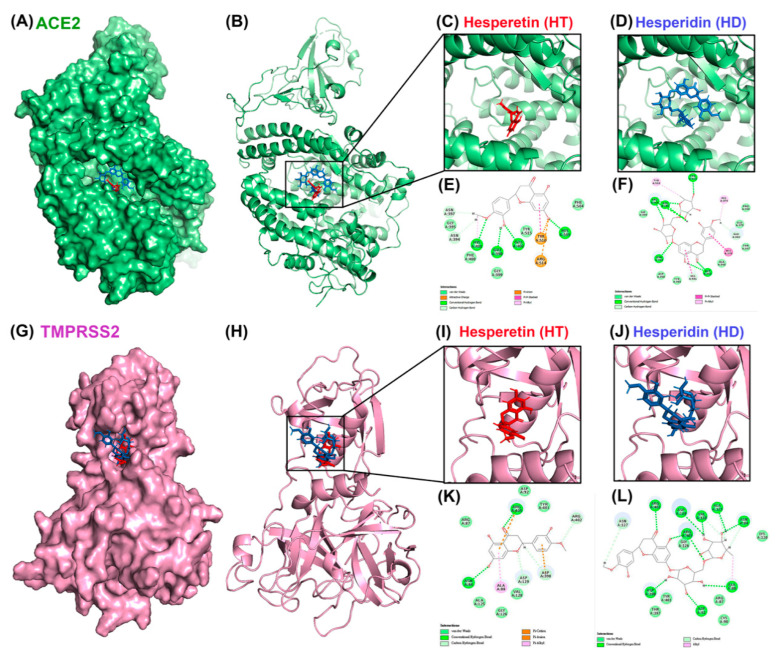
Molecular docking pose visualization for the interaction of ACE2/TMPRSS2 and HT/HD. Compound–protein interaction between HT (PDB code: 5JDC; red structure)/HD (PDB code: 6CCF; blue structure) and ACE2 protein (PDB code: 3D0G) in 3D (**A**–**D**) and 2D (**E**,**F**). Compound–protein interaction between HT/HD and TMPRSS2 protein (PDB code: 1Z8A) in 3D (**G**–**J**) and 2D (**K**,**L**).

**Figure 2 nutrients-13-02800-f002:**
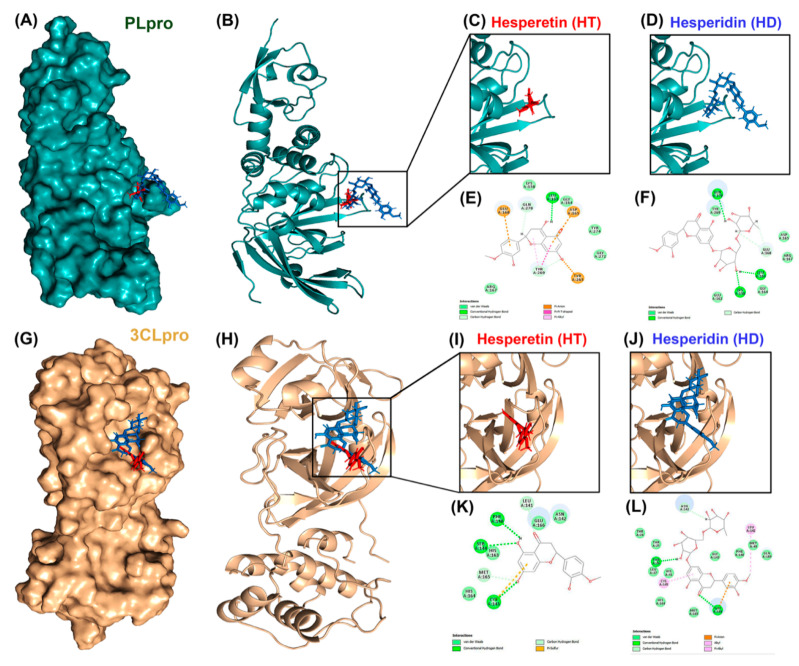
Molecular docking pose visualization for the interaction of PLpro/3CLpro and HT/HD. Compound–protein interaction between HT (PDB code: 5JDC; red)/HD (PDB code: 6CCF; blue) and PLpro protein (PDB code: 3E9S) in 3D (**A**–**D**) and 2D (**E**,**F**). Compound–protein interaction between HT/HD and 3CLpro protein (PDB code: 3AW0) in 3D (**G**–**J**) and 2D (**K**,**L**).

**Figure 3 nutrients-13-02800-f003:**
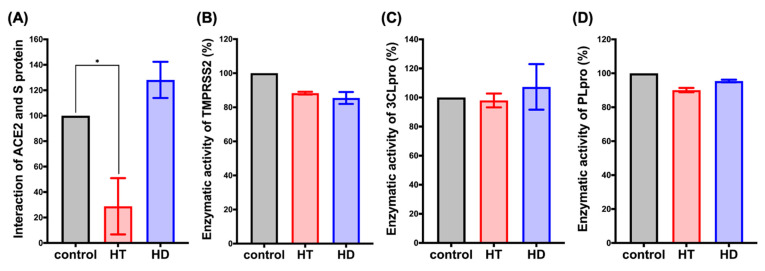
HT (Hesperetin) decreased the interaction of ACE2 and the spike protein. FRET assay was performed to determine the interaction between human receptor ACE2 and S protein (**A**). The in vitro enzymatic activity of TMPRSS2 (**B**), PLpro (**C**), and 3CLpro (**D**) was determined after 1 hr incubation with HT and HD (Hesperidin). Data are shown as mean ± SEM from 3 independent experiments with triplicates. * *p* < 0.05.

**Figure 4 nutrients-13-02800-f004:**
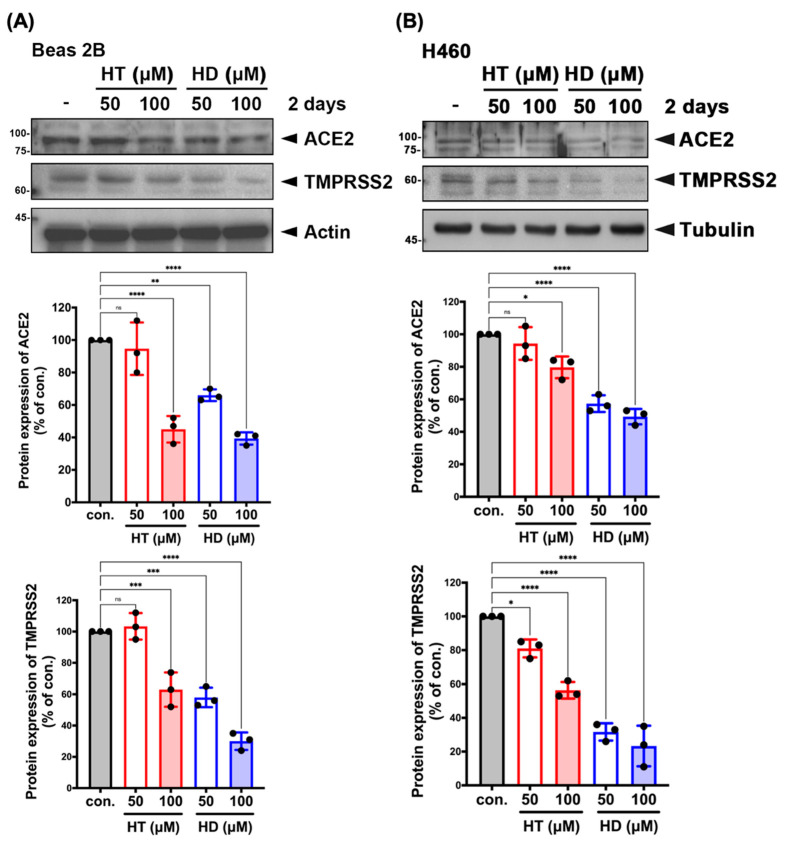
HT (Hesperetin) and HD (Hesperidin) suppressed the protein expressions of ACE2 and TMPRSS2 in normal and malignant lung cells. Beas 2B (**A**) and H460 (**B**) cell lines were treated with HT and HD in a dose-dependent manner for 2 days followed by the examination of protein expression in Western blotting with indicated antibodies. The quantitative results of ACE2 and TMPRSS2 expressions in Beas 2B and H460 were normalized with a level of actin and were shown below to Western blot images. *●* is shown as the mean of every independent experiment. Data are shown as mean ± SEM from 3 independent experiments with triplicates. * *p* < 0.05, ** *p* < 0.01, *** *p* < 0.001, and **** *p* < 0.0001. NS, no significance.

**Figure 5 nutrients-13-02800-f005:**
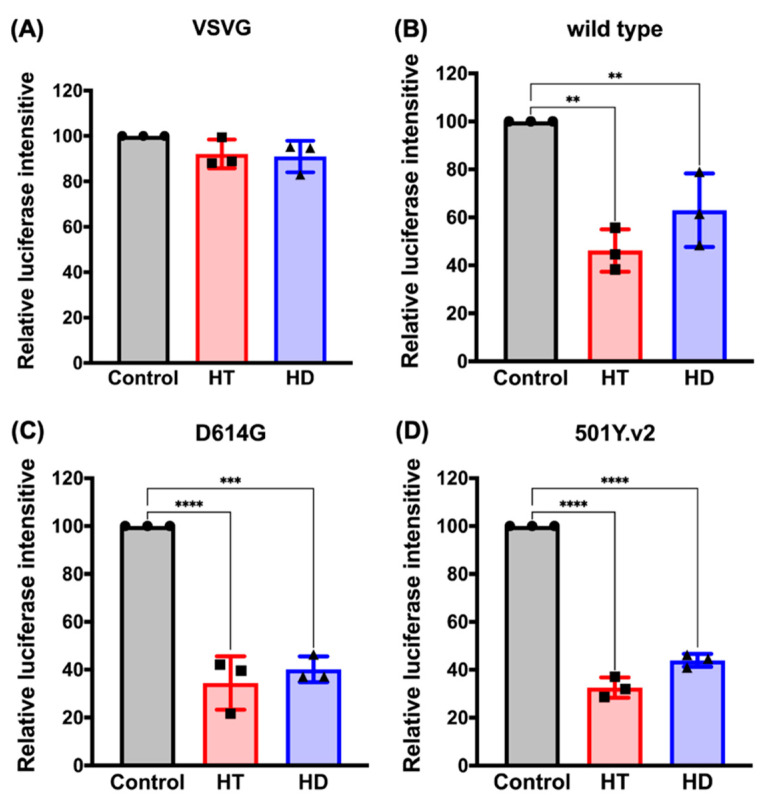
HT (Hesperetin) and HD (Hesperidin) impede SARS-CoV-2 pseudovirus into VeroE6 cell. The luciferase intensity of VSVG (**A**) or SARS-CoV-2 pseudovirus with S protein wild type (**B**), D614G strain (**C**), and 501Y.v2 stain (**D**) in VeroE6 cell line was measured after treatments with 100 μM of HT and HD for 2 days. ●, ■, and ▲ are shown as the mean of every independent experiment. Data are shown as mean ± SEM from 3 independent experiments with triplicates. ** *p* < 0.01, *** *p* < 0.001, and **** *p* < 0.0001.

**Table 1 nutrients-13-02800-t001:** The best predicted energy values of hesperetin and hesperidin with proteins related to SARS-CoV2.

		c-Docker Energy Value (kcal/mol)	Residues	Distance (Å)	Types
ACE2	HT	−34.81	Ala348	1.9	H-bond (H34, O)
			Asp382	2.76	CH-bond (H31, OD1)
			Glu398	2.42	H-bond (H36, O)
				2.44	CH-bond (HA, O21)
			Glu402	2.23	H-bond (HN, O20)
	HD	−1.65	Asp206	2.97/2.97	CH-bond (H52, OD2)/(H54, OD2)
			Thr347	2.74	CH-bond (HA, O14)
			Ala348	2.66	H-bond (H73, O)
			Glu375	2.02/2.17	H-bond (H74, OE1)/(H75, OE1)
			Asp382	2.09	H-bond (HD2, O9)
				2.59	CH-bond (H57, OD1)
			Glu398	2.57	CH-bond (H52, OE1)
			His401	2.76	p-s
			Glu402	2.69	CH-bond (H65, OE1)
			Arg514	4.78	p-cation
TMPRSS2	HT	−30.56	Lys254	3.92	p-cation
			Gly378	2.15	H-bond (H67, O)
	HD	−7.2	His203	4.81	p-alkyl
			Lys254	1.94	H-bond (HZ1, O15)
				1.75	H-bond (HZ3, O14)
				4.08	p-cation
				2.65	CH-bond (HE1, O6)
			Glu301	2.68/2.74	H-bond
					(H72, O)/(H71, OE2)
				2.47	CH-bond (H56, OE2)
PLpro	HT	−13.69	Asp165	2.98	CH-bond (H31, OD2)
			Pro248	4.97	p-alkyl
	HD	17	Leu163	2.28	H-bond (H77, O)
			Gly164	5.21	Amide-π stracked
			Asp165	2.96	p-anion
			Glu168	2.94	H-bond (H71, OE2)
			Pro249	2.64	CH-bond (HD2, O6)
			Tyr265	5.47	p-p T shaped
			Gln270	2.44	CH-bond (H60, OE1)
				2.31	Unfavorable donor-donor
					(HE22, H75)
3CLpro	HT	−17.94	Phe140	1.94	H-bond (H34, O)
			Asn142	2.96	H-bond (HD21, O18)
			Glu166	2.75	H-bond (H34, OE1)
				3.07	p-cation
			Gln189	2.19	H-bond (HE22, O21)
	HD	6.21	His41	2.34	H-bond (HE2, O15)
				4.6	p-alkyl
			Asn142	2.63	CH-bond (H46, OD1)
				2.81/2.85	H-bond (HN, O2)/(HD21, O3)
			Gly143	2.8	H-bond (HN, O14)
			Cys145	2.27	H-bond (HG, O15)
			Met165	5.17	Alkyl
			Glu166	2.47/2.69	CH-bond (H61, O)/(H59, O)
				1.95	H-bond (H73, O)

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
