# Peer review of "Hesperidin Is a Potential Inhibitor against SARS-CoV-2 Infection"

_nutrients, 2021, doi:10.3390/nu13082800_

Round 1
Reviewer 1 Report
In general, research devoted to drug repurposing strategy and to find new potential antiviral agents against SARS-CoV-2 virus are relevant to current needs and could be interesting as well as valuable. This can significantly reduce the time and resources required to advance a candidate antiviral drug into the clinic. These advantages are particularly relevant for emerging viral diseases. In this study, Cheng et al., found that hesperidin (HD) and its aglycone metabolite “Hesperetin (HT)” can inhibit the replication of SARS-CoV-2 in Vero E6 cells via blocking the interaction between spike protein and cellular receptor ACE2 and reducing ACE2 and TMPRSS2 expression, at early stage of viral infection. Overall, the antiviral activity was well assessed in this study, however
- These compounds have already been known as exhibiting antiviral activity. The work supposed to present novelty since the authors hypothesize that all these compounds are inhibitors of cellular ACE2 and TMPRSS2 expression but did not discuss viral targets like virus main protease Mpro (e.g. https://doi.org/10.3390/pathogens10060758).
- The authors must provide quantitative calculations of Figure 4 in bars or curves with their corresponding western blots to be able to clearly present the impact on TMPRSS2 expression.
Author Response
We thank this reviewer for the important comments and suggestions. The questions were accordingly addressed point-by-point as follows. Please check the attached file as the rebuttal letter.

Reviewer 2 Report
Dear Authors,
first of all, I thank You for giving me the opportunity to read this article.
You proposed hesperidin as a potential inhibitor against SARS-CoV-2 infection.
Unfortunately, the article was written in a confused way. For instance, lines from 65 to 68 should be removed because were repetitive; lines from 71 to 76 should be removed because You did not have to anticipate conclusions in the Introduction section; and so on.....
In addition to this, the research seemed not conducted correctly, because methodological forcing was present .
Finally, the discussion section was too speculative.
Author Response
We thank this reviewer for the important comments and suggestions. Please check the attached file as the rebuttal letter.
Reviewer #2
Dear Authors,
first of all, I thank You for giving me the opportunity to read this article. You proposed hesperidin as a potential inhibitor against SARS-CoV-2 infection. Unfortunately, the article was written in a confused way. For instance, lines from 65 to 68 should be removed because were repetitive; lines from 71 to 76 should be removed because You did not have to anticipate conclusions in the Introduction section; and so on..... In addition to this, the research seemed not conducted correctly, because methodological forcing was present. Finally, the discussion section was too speculative.
Authors’ Response:
We thank this reviewer for the critical comments. Per the review’s suggestion, we removed the repetitive description and anticipate conclusion in the Introduction section. The writing in the Introduction and Discussion sections was also revised. Some data, such as Figure 4, has been improved according to the reviewers’ suggestions

Reviewer 3 Report
In this study entitled “Hesperidin is a potential inhibitor against SARS-CoV-2 infection” by Cheng et al, the authors identified the flavanone glycoside Hesperedin (HD) and its aglycone metabolite Hesperetin (HT) as compounds binding to the ACE2 and TMPRSS2, two key cellular proteins involved in SARS-CoV-2 entry into cells. Using a TR-FRET assay the authors demonstrate potential effects of HD on Spike-ACE2 interaction and using a SARS-CoV-2 spike pseudotyped particle system, the authors show reduced viral entry in presence of these compounds. They also show potential depletion of ACE2 and TMPRSS2 in cells treated with HD and HT. While the concept of the study is interesting and novel, the experimental techniques used have several critical flaws which prevent drawing firm conclusions regarding the presented data. The reviewer recommends major changes to the study, to bring the data quality up to publication standards.
The major recommendations are
- The data in figures 3-5 are from experiments performed in triplicate, according to the figure legends. The triplicates can only be considered technical repetitions and not biological repetitions. The authors must perform a minimum of three independent biological repeats for each experiment to draw valid conclusions from these experiments. The authors should use ANOVA instead of T-test to check whether there is significant difference between HT and HD in various test conditions.
- In Fig.3a, the TR-FRET assay is lacking controls to show that the drop in signal observed with HT is not due to its direct effect on fluorescence of the fluorophores conjugated to ACE2 and Spike protein. The authors should show data of those control samples. The authors also must show a dose response curve and calculate the IC50 for HT.
- The authors must use another complimentary method like microscale thermophoresis or surface plasmon resonance (biacore) to show that HT can indeed destabilize spike-ACE2 interaction.
- In Fig. 4, the quality of western blot signal for ACE2 and TMPRSS2 is poor and needs improvement. The authors also provide quantitation and statistical analysis of the western blot and should include another marker like GAPDH to show that the observed effects are not due to global effects on transcription and translation. The mRNA transcript level of both proteins (ACE2 and TMPRSS2) must be measured to see whether there is reduction in transcription of these proteins. The authors have not provided data on potential mechanisms behind the depletion of the two proteins. The authors should check whether there is an accelerated degradation of these proteins using proteasomal and autophagy inhibitors.
- Figure 5 does not show the data on how VSVG pseudotyped particles behave in presence of the compounds HT and HD (VSVG is included as control in the kit the authors are suing to measure viral entry). This is a vital control that need to be included as part of the data.
- The current experiment format for fig. 5 pre-treat the cells with the compounds for 2 days before infecting them with pseudotyped lentiviruses. What is the rationale behind this? If the compounds act by blocking spike’s interaction with ACE2 or TMPRSS2, pre-treatment is unnecessary, and the compounds can be added either to the virus stock or added along with the virus during infection. Adding an additional control of adding the compounds 2 hours post-infection would reveal whether the compounds act at a different step which is not relevant in context of SARS-CoV-2 infection.
- The mode of action of compound HD is a bit of mystery as it did not block ACE2-Spike interaction (fig. 3) or had any effects on viral proteases. The authors must provide a better explanation on how they think this compound is working.
- For confirmation that these compounds have potential to be developed to antivirals, the authors must include experiments with authentic SARS-CoV-2 infection, ideally in a primary cell context or an animal model (if possible).
Author Response

(The authors gave the same response as above.)

Round 2
Reviewer 2 Report
Dear Authors,
I read the revised version of Your manuscript.
All my comments and suggestions were satisfatorily met.
Reviewer 3 Report
The authors have satisfactlrily addressed most of the concerns raised by this reviewer. I think the manuscript is now ready for publication in 'Nutrients'.